# Tadalafil 5 mg Alone or in Combination with Tamsulosin 0.4 mg for the Management of Men with Lower Urinary Tract Symptoms and Erectile Dysfunction: Results of a Prospective Observational Trial

**DOI:** 10.3390/jcm8081126

**Published:** 2019-07-29

**Authors:** Arcangelo Sebastianelli, Pietro Spatafora, Jacopo Frizzi, Omar Saleh, Maurizio Sessa, Cosimo De Nunzio, Andrea Tubaro, Linda Vignozzi, Mario Maggi, Sergio Serni, Kevin T. McVary, Steven A. Kaplan, Stavros Gravas, Christopher Chapple, Mauro Gacci

**Affiliations:** 1Department of Minimally Invasive and Robotic Urologic Surgery and Kidney Transplantation, University of Florence, 50134 Florence, Italy; 2Department of Experimental Medicine, Section of Pharmacology “L. Donatelli”, University of Campania “L. Vanvitelli”, 80138 Naples, Italy; 3Department of Urology, Sant’Andrea Hospital, University “La Sapienza”, 00189 Rome, Italy; 4Department of Clinical Physiopathology, University of Florence, 50134 Florence, Italy; 5Division of Urology, Southern Illinois University School of Medicine, Springfield, IL 62702, USA; 6Department of Urology, Icahn School of Medicine at Mount Sinai, New York City, New York, NY 10029, USA; 7Department of Urology, University of Thessaly, Larissa 41222, Greece; 8Department of Urology, Sheffield Teaching Hospitals NHS Trust, Sheffield S102JF, UK

**Keywords:** benign prostatic hyperplasia, erectile dysfunction, lower urinary tract symptoms, medical therapy, tadalafil, tamsulosin

## Abstract

Tadalafil 5 mg represents the standard for men with Erectile dysfunction (ED) and lower urinary tract symptoms (LUTS)/benign prostatic enlargement (BPE). We carried out an observational trial aiming to assess the efficacy and safety of Tadalafil compared with Tadalafil plus Tamsulosin. Seventy-five patients complaining of ED and LUTS were treated for 12-weeks with Tadalafil plus placebo (TAD+PLA-group) or with combination therapy tadalafil plus tamsulosin (TAD+TAM-group). Efficacy variables were: International Index of Erectile Function (IIEF), International Prostate Symptom Score (IPSS), maximum urinary flow rate (Qmax) and safety assessments. Data were evaluated using paired samples *T*-test (baseline vs. 12-weeks) and analysis of variance (Δgroup-TAD+PLA vs. Δgroup-TAD+TAM). At baseline, both groups presented similar characteristics and symptoms scores (all: *p* > 0.05). From baseline to 12-weeks, all the subjects showed a significant improvement of IIEF, total-IPSS, storage-IPSS, Qmax (all: *p* < 0.001). Conversely, a significant improvement of voiding-IPSS was observed in TAD+TAM-group (−3.5 points, *p* < 0.001). Indeed, TAD+PLA-group showed a not significant improvement of voiding-IPSS (−2.0 points, *p* = 0.074). When we compared between-groups differences at 12-weeks, IIEF (*p* = 0.255), total-IPSS (*p* = 0.084) and storage-IPSS (*p* = 0.08) did not show any statistically significant differences, whereas, voiding-IPSS and Qmax were significantly better in TAD+TAM-group (*p* = 0.006 and *p* = 0.027, respectively). No severe treatment adverse events (TAEs) were reported in both groups. Tadalafil achieved the same improvements of IIEF, total-IPSS, storage-IPSS when compared to combination therapy. Instead, Qmax and voiding-IPSS were better managed with combination therapy, without change of TAEs.

## 1. Introduction

Erectile dysfunction (ED) and lower urinary tract symptoms (LUTS) related to benign prostatic obstruction (BPO) are common conditions in middle-age or older men [1]. Men with LUTS have a higher incidence of ED, and LUTS themselves represent an independent risk factor for ED, [2] triggering a significant negative impact on quality of life (QoL). The underlying pathophysiological links between LUTS secondary to benign prostatic hyperplasia (BPH) and ED are not yet completely understood, even though several determinants are shared by these two clinical entities [1,3].

Alpha-blockers (ABs) and 5a-reductase inhibitors (5 ARIs), alone or in combination, are the mostly prescribed drugs to manage patients with LUTS/BPH [4]. ABs are usually the first line treatment for LUTS thanks to their rapid onset of action. By antagonizing alpha(1A)-adrenergic receptors in the prostate and urethra, they cause smooth muscle relaxation in lower urinary tract (LUT) determining the decrease of the functional obstruction [5,6].

The association of AB with 5 ARI has been proposed and widely adopted in order to reduce the overall risk of BPH progression and to improve LUTS in patients with moderate enlarged (≥40 mL) prostates when long term treatment is planned. However, in several sub analysis voiding symptoms and maximum urinary flow rate (Qmax) improvement was not as high as expected. Moreover, these therapies have potential major side-effects on sexual activity, particularly in erectile and ejaculatory function [7,8,9].

Phosphodiesterase type 5 inhibitors (PDE5-Is) were approved for the treatment of ED, but they have been also demonstrated as highly effective in blunting LUTS in men with or without ED [10,11,12,13]. Indeed, PDE5-Is act on the relaxation of bladder neck and prostate by increasing nitric oxide in smooth muscle, allowing a direct action on micturition phases and not only penile erection. [14,15].

Moreover, they exert potent anti-inflammatory effects on prostate therefore reducing fibrosis and overgrowth. All these beneficial effects help in maintaining prostatic structural anatomy and physiological activity [16,17].

Currently the efficacy of PDE5-Is on LUTS recovery has been well established, and tadalafil 5 mg once daily has been approved for the treatment of LUTS/BPH with or without coexisting ED [3].

Recently, a meta-analysis of 12 randomized controlled trials (RCTs) on PDE5-Is suggested that combination therapy with ABs, particularly with tamsulosin, the only AB approved by the Food and Drug Administration in combination with tadalafil for the treatment of LUTS, allows achieving an additional significant improvement of International Prostate Symptom Score (IPSS) as compared to ABs alone. Conversely, we have no definitive evidence for the comparison between PDE5-Is plus ABs vs. PDE5-Is alone [18].

We carried out a prospective placebo-controlled trial, with the aim to compare the efficacy and safety of daily administration tadalafil 5 mg alone (tadalafil plus placebo) with the combination tadalafil 5 mg plus tamsulosin 0.4 mg in men with LUTS/BPH and ED.

## 2. Materials and Methods

### 2.1. Study Population and Design

Our prospective observational trial was conducted in two departments of urology. Across a period of 12 months, 82 consecutive men presenting with ED and LUTS suggestive of BPO were enrolled.

Inclusion criteria were: age >40 to 80 years, mild to severe ED (International Index of Erectile Function–Erectile Function-5 <22), moderate to severe LUTS (International Prostate Symptom Score >7) and with Qmax >5 mL/s obtained from a uroflowmetry assessment. Exclusion criteria were: hypersensitivity to tadalafil or tamsulosin, prostatic cancer or suspected with prostate-specific antigen (PSA) >4 ng/mL, bladder lithiasis, previous prostatic surgery, urinary tract infection, neurogenic bladder, finasteride or dutasteride use within 3 or 6 months, respectively, clinical history of urethral and/or proven bladder neck obstruction.

The present trial was carried out according to the ethical principles of the Declaration of Helsinki, following the rules of good clinical practices. The study was approved by the review boards of the two involved institutions (Ethics Committee Approval OSS.15.031, approved on 22 June 2015). All men provided written informed consent before initiating any trial procedure or therapy.

The assessment of patients included age, body mass index (BMI), waist circumference (WC), blood pressure, clinical laboratory parameters, digital rectal examination. LUTS and ED were assessed using validated questionnaires. In particular, LUTS with total International Prostate Symptom Score (IPSS), focusing also on storage and voiding IPSS subscores, nocturia question and IPSS QoL [19], ED with International Index of Erectile Function-5 (IIEF-5) [20]. Uroflowmetry was performed by all the subjects enrolled and postvoid residual volume (PVR) was evaluated with abdominal ultrasound immediately after voiding.

Subjects reporting intake of BPH, overactive bladder, or ED therapies underwent a 4 weeks treatment-free washout period. Moreover, all subjects enrolled underwent a 2 weeks run-in period with tamsulosin 0.4 mg/day alone. Fifty patients enrolled in one center, were treated with tadalafil 5 mg/day plus tamsulosin 0.4 mg/day (TAD+TAM-group) for 12 weeks, otherwise 25 patients enrolled in the other center received tadalafil 5 mg/day plus placebo (TAD+PLA-group). The medications were self-administered every day at the same time, before the night rest, without any limitations or variations of sexual activity timing or food intake. Patients were evaluated at screening, run-in, baseline, and after 12 weeks of treatment (Figure 1).

Safety was assessed by evaluating subject-reported adverse events (AEs), orthostatic vital signs, clinical laboratory parameters, uroflowmetry and PVR.

Patients with incomplete data sets were excluded from statistical analysis.

### 2.2. Statistical Analysis

Differences between tadalafil monotherapy (TAD+PLA-group) and combination therapy (p TAD+TAM-group) were calculated at baseline and 12 weeks, by using the unpaired sample *t*-test. Instead, the paired sample *t*-test was adopted to assess mean changes between baseline and week 12 visit in both groups. The one-way analysis of variance (ANOVA) was used to measure the between-group changes from baseline to 12 weeks. All statistical analyses were done with SPSS® (SPSS Inc., Chicago, IL, USA). A *p* value of 0.05 or less was considered statistically significant.

## 3. Results

### Experimental Results

A Consolidated Standards of Reporting Trials (CONSORT) flow chart is shown in Figure 1. Of 82 subjects enrolled, 75 were screened. All subjects completed the study. Both groups showed a >90% compliance with dosing requirements.

Mean age was 65 years. No significantly differences in baseline characteristics, including IPSS, and the other BPH-related characteristics (Qmax, symptoms score), were observed between the two groups (Table 1). After 12 weeks of treatment from baseline, total IPSS, storage IPSS, IPSS QoL, IIEF-5 and Qmax statistically significantly improved in both groups (all: *p* < 0.001) (Table 2). Nevertheless, only the subjects in TAD+TAM-group, treated with combination therapy, experienced a statistically and clinically significant recovery of voiding LUTS, with a reduction of −3.5 voiding IPSS points (*p* < 0.001), when compared with tadalafil monotherapy (−2.0 voiding IPSS points, *p* = 0.074) (Table 2). No significant differences between TAD+TAM-group and TAD+PLA-group, except for Qmax (*p* = 0.027) and voiding IPSS (*p* = 0.006), emerged at between-group ANOVA analysis at week 12 endpoint (Figure 2).

The proportion of subjects reporting at least one treatment-emergent AE (TEAE) was 16% in TAD+PLA-group and 22% in TAD+TAM-group (Table 3). TEAEs were mild to moderate in severity. The most common TEAEs in TAD+PLA-group were headache (*n* = 2) followed by back pain (*n* = 1) and nasopharyngitis (*n* = 1), whereas the most common TEAEs with combination therapy (TAD+TAM) were headache (*n* = 4), back pain (*n* = 3), ejaculatory dysfunction (semen volume decreased/retrograde ejaculation) (*n* = 2), dyspepsia (*n* = 1) and dizziness (*n* = 1). Laboratory measurements or vital signs did not show any clinically significant changes. No episodes of urinary retention were reported. No subject discontinued therapy because of TEAEs.

## 4. Discussion

In this study, we found that daily administration of tadalafil 5 mg was effective in improving the overall urinary symptoms, without any difference between monotherapy or combination with tamsulosin 0.4 mg, but also uroflowmetry parameters. Nevertheless, men treated with combination therapy showed a more remarkable improvement of Qmax and voiding IPSS compared with tadalafil alone.

In elderly men ED and LUTS related to benign prostatic enlargement (BPE) represent a high prevalence comorbid condition with a negative impact on patients’ QoL and significant economic burden. It has been established in preclinical and clinical trials that besides aging, several metabolic factors affect the onset and worsening of both ED and LUTS, concurring to penile and nerves alterations and also prostate enlargement and inflammation [21]. Even if the pathophysiological pathways shared between ED and LUTS are still not totally elucidated, PDE5-Is proved to be effective for the treatment of both these conditions [1,14].

Indeed, PDE5-Is are able to manage prostate inflammation and could act on the related fibrosis by improving pelvic and prostate oxygenation. Moreover, it seems that they might contribute to restore the physiologic activity of prostate and to stabilize the glandular structural anatomy [17,22,23].

Other relevant PDE5-Is’ mechanisms of action are emerging and are currently under investigation. There is a significant body of evidence supporting the relaxating action of PDE5-Is on the smooth muscle fibers of bladder neck, urethra, and prostate. Furtehrmore, even if with less effectiveness, they might reduce the tone of bladder muscles and affect the micturition reflex, by improving LUT blood supply and modulating bladder afferent innervation [14].

Several RCTs demonstrated that PDE5-Is are able to significantly decrease IPSS score, ameliorating both storage and voiding LUTS, and improve patients’ QoL. However, in most trials, Qmax was not significantly different from placebo [10,24,25,26].

Likewise, in a meta-analysis by Gacci et al., IPSS and IIEF scores, but not Qmax, were significantly improved by PDE5-Is, [14,27] as in the first systematic review about the use of PDE5-Is for LUTS associated with BPE, Laydner et al. reported that PDE5-Is improve IPSS score and IIEF-5 but not Qmax [28]. However, in a recent study by Roehrborn et al. evaluating the efficacy of tadalafil 5 mg once daily, a slight but significant Qmax increase was observed [26].

Moreover, in a subset analysis based on data from a systematic review, daily administration of tadalafil 5 mg was associated with a remarkable improvement of both ED and LUTS/BPH. [29]. Thus, according to EAU guidelines, tadalafil 5 mg is presently considered a valuable treatment option also for men with moderate to severe LUTS suggestive of BPE [4].

In clinical practice, tadalafil once daily is increasingly prescribed as first-line therapy for LUTS/BPE and concomitant ED. However, several patients, in particular those with prevalent voiding LUTS, often switch to other medical treatments, rarely considering an ongoing tadalafil/tamsulosin combination therapy, even if combination therapy has been proposed by several authors [30,31].

From our results, both subjective and objective parameters were significantly improved at the end of the trial in the 2 treatment arms, supporting the evidence for the use of tadalafil 5 mg as monotherapy or in combination with tamsulosin 0.4 mg in men with ED and LUTS. In particular, we observed a clinically meaningful recovery of LUTS, since a decrease ≥25% or ≥3 points of total IPSS was achieved in both groups. However, at the end of the trial, Qmax and voiding IPSS were significantly better in men treated with combination therapy compared to tadalafil only. Indeed, tadalafil 5 mg significantly decreased total IPSS of >30% and improved Qmax after 12 weeks of monotherapy (mean improvement of Qmax: +2.24 mL/s). Nevertheless, higher improvements in voiding symptoms were observed with combination of tadalafil and tamsulosin.

The results from the first meta-analysis on PDE5-Is for the treatment of ED and LUTS, proved the improvement of IPSS score and Qmax, besides ED evaluated with IIEF score, in men treated with the association of ABs and PDE5-Is compared to ABs alone. [18]. Our data support and enhance the foundings of a recent meta-analysis by Yan et al. Indeed, compared with the use of PDE5-Is alone, the combined therapy of PDE5-Is plus ABs allowed achieving a significant improvement of sexual activity (+2.25 mean difference of IIEF) and LUTS (−4.21 of IPSS), (+1.43 in Qmax) [32].

Accordingly, the results of our study suggest that adding tamsulosin to tadalafil allows to achieve a further improvement of Qmax and voiding symptoms as compared to tadalafil monotherapy after 12 weeks of therapy. However, the similar overall improvement of LUTS (total IPSS and IPSS QoL) and ED (IIEF-5) between the 2 treatment arms, may theoretically allow a increasingly patient-oriented personalized therapy.

Concerning Qmax, McVary et al. demonstrated in two 12 weeks randomized studies comparing tadalafil or sildenafil with placebo, a nearly double improvement of Qmax in subjects treated with PDE5-Is. [10,33].

Anyway, as reported by Tuncel et al., the improvement of Qmax was significantly better in patient treated with combination therapy (+42%) of sildenafil and tamsulosin, compared with the two drugs alone [34]. 

Similar findings have been showed by Kaplan et al. in a randomized trial comparing alfuzosin with sildenafil or the combination of both. After 12 weeks the improvement of Qmax was 21% for combination therapy, 6% for sildenafil and 11% for alfuzosin [35].

Our results, in agreement with the previous studies, [10,33,34,35] proved that voiding LUTS and uroflowmetric parameters could be significantly further improved with the addition ABs to PDE5 I therapy. Thus, based on prevailing LUTS reported at first or at follow-up visit, a more precise patients’ counselling and medications might be offered in daily clinical practice, according to our findings. Tadalafil 5 mg daily dose was well tolerated both in monotherapy or in combination with tamsulosin 0.4 mg.

Even if the occurrence of AEs was slightly higher in combination therapy group (22% vs. 16%), none of them was severe and we did not observe any withdrawal from the trial due to AEs. Conversely Bechara et al. reported a 1/15 discontinuation in the group treated with tadalafil plus tamsulosin [36] and Chung et al. 4/82 discontinuations [37] with tamsulosin associated with udenafil. Headache was the main AE reported, and the same incidence in both groups (8%) was found. Moreover, in line with previous data, the co-administration of tadalafil and tamsulosin was not associated with symptomatic hypotension [38].

The strengths of the present study are the well sized and homogenous population, the prospective organization of the trial and the data collection. However, this study may be restricted by several limitations. Because this study was observational, it could be prone to biases. Moreover, we did not consider prostate size as inclusion or exclusion criterion or as a possible factor influencing the effectiveness of treatments evaluated. A long-term follow-up is also needed.

## 5. Conclusions

Tadalafil 5 mg daily monotherapy is able to improve ED and overall LUTS (IPSS and Qmax) after 12 weeks. However, the addition of tamsulosin 0.4 mg to tadalafil 5 mg can further enhance the improvement of voiding symptoms and Qmax. Combination therapy is well tolerated. Even if the overall occurrence of ds is slightly higher as compared with tadalafil alone, their low severity allows achieving good compliance and safety.

Further studies, with longer follow-ups, are still required to assess long term efficacy and safety of tadalafil and tamsulosin coadministration. 

## Figures and Tables

**Figure 1 jcm-08-01126-f001:**
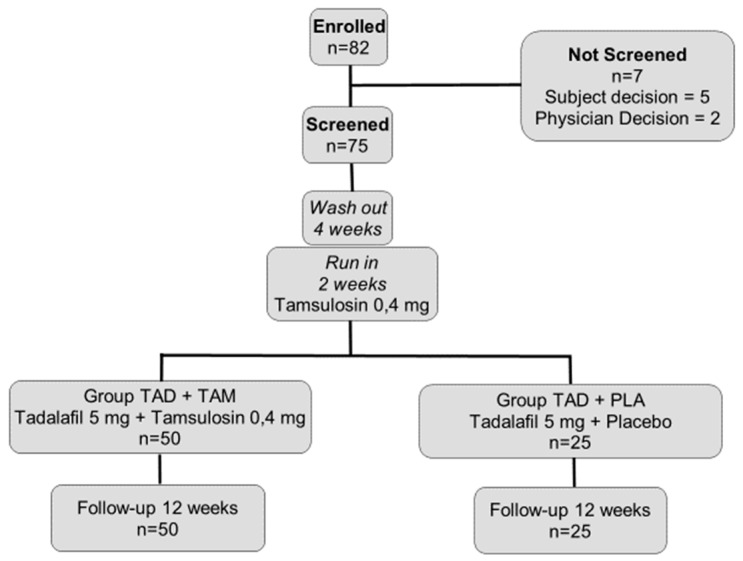
Disposition of subjects according to Subject Consolidated Standards of Reporting Trials (CONSORT) diagram.

**Figure 2 jcm-08-01126-f002:**
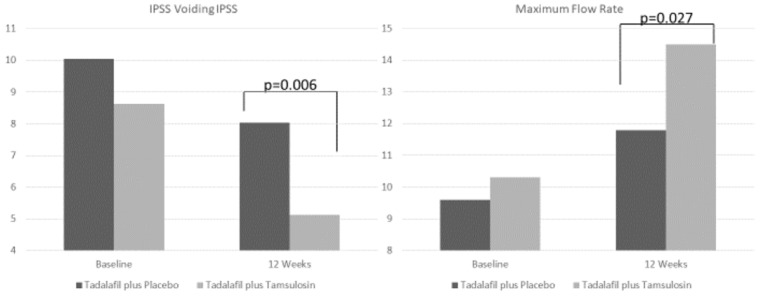
Change from baseline to 12 weeks for tadalafil and tadalafil plus tamsulosin (combination) groups. Graphical representation of the significant differences between the two groups. IPSS = International Prostate Symptom Score.

**Table 1 jcm-08-01126-t001:** Patients’ baseline characteristics.

All Patients	TAD + TAM Group *n* = 50		TAD + PLA Group *n* = 25		*p*
Baseline Characteristics	Mean ± SD Deviation	(Minimum-Maximum)	Mean ± SD Deviation	(Minimum-Maximum)	
Age (years)	65.7 ± 9.1	47–78	65.5 ± 6.3	51–74	0.238
Weight (kg)	78.2 ± 9.2	69–86	75.8 ± 10.4	67–83	0.216
Body mass index (kg/m2)	27 ± 3.1	25–31	26.7 ± 3.6	24–31	0.174
Abdominal obesity: waist circumference (cm)	108.8 ± 4.1	92–135	102.3 ± 5.4	76–120	0.136
Triglycerides (mg/dL)	156.7 ± 8.4	76–247	129.2 ± 7.2	83–185	0.117
HDL cholesterol (mg/dL)	49.4 ± 2.8	31–76	49.8 ± 2.3	32–60	0.259
Glycemia (mg/dL)	111.3 ± 3.5	76–211	102.7 ± 5.7	72–188	0.113
IPSS base	18.8 ± 5.9	8–32	17 ± 6.1	8–29	0.224
IPSS voiding base	8.6 ± 3.8	1–20	10 ± 4.1	3–18	0.146
IPSS storage base	8.3 ± 3.2	0–14	6.9 ± 4.2	1–15	0.118
IPSS QoL base	3.9 ± 1	2–6	3.5 ± 1.4	1–6	0.145
IIEF-5 base	12 ± 3.5	6–21	13.8 ± 5.2	1–21	0.09
Q max base (mL/s)	10.3 ± 3.5	3.4–17.8	9.6 ± 2.8	6–17	0.369

SD = standard deviation; IPSS = International Prostate Symptom Score; IIEF-5 = International Index of Erectile Function-5; Qmax = maximum urinary flow rate.

**Table 2 jcm-08-01126-t002:** Change from baseline to 12 weeks in tadalafil monotherapy (TAD+PLA-Group) and tadalafil plus tamsulosin (TAD+TAM-Group) groups.

Variables Assesed	TAD + TAM Group *n* = 50	TAD + PLA Group *n* = 25	*p* Value (Anova Analysis)
IPSS 12 week (Mean ± SD)	11.5 ± 5.4	11.8 ± 6.3	
Delta3 M (baseline - 12wks)	−7	−5.2	
*p* value (paired samples *T*-test)	<0.001	<0.001	0.084
IPSS voiding 12 week (Mean ± SD)	5.1 ± 2.7	8 ± 4.7	
Delta3 M (baseline - 12wks)	−3.5	−2	
*p* value (paired samples *T*-test)	<0.001	0.074	0.006
IPSS storage 12 week (Mean ± SD)	5.3 ± 2.7	3.8 ± 3.4	
Delta3 M (baseline - 12wks)	−3	−3.1	
*p* value (paired samples *T*-test)	<0.001	<0.001	0.08
IPSS QoL 12 week (Mean ± SD)	2.1 ± 1	2.1 ± 1.7	
Delta3 M (baseline - 12wks)	−1.8	−1.3	
*p* value (paired samples *T*-test)	<0.001	0.009	0.321
IIEF-5 12 week (Mean ± SD)	17.7 ± 3.3	19.9 ± 5.1	
Delta3 M (baseline - 12wks)	5.7	6.1	
*p* value (paired samples *T*-test)	<0.001	<0.001	0.255
Q max 12 week (Mean ± SD)	14.5 ± 3.7	11.8 ± 4	
Delta3 M (baseline - 12wks)	4.2	2.2	
*p* value (paired samples *T*-test)	<0.001	<0.001	0.027

*p* value: significance of the analysis of variance (ANOVA). SD = standard deviation; IPSS = International Prostate Symptom Score; IIEF-5 = International Index of Erectile Function-5; Qmax = maximum urinary flow rate.

**Table 3 jcm-08-01126-t003:** Summary of adverse events at treatment period (12 weeks).

Adverse Events	TAD + TAM Group *n* = 50, (*n* %)	TAD + PLA Group *n* = 25, (*n* %)	*p* Value
Any TEAEs	11 (22%)	4 (16%)	0.075
Serious AEs	0 (0%)	0 (0%)	0.267
Intensity			
mild	7 (14%)	3 (12%)	0.114
moderate	4 (8%)	1 (4%)	0.098
severe	0 (0%)	0 (0%)	0.286
Headache	4 (8%)	2 (8%)	0.163
Nasopharyngitis	0 (0%)	1 (4%)	0.196
Back pain	3 (6%)	1(4%)	0.087
Dizziness	1 (2%)	0 (0%)	0.173
Dyspepsia	1 (2%)	0 (0%)	0.185
Ejaculatory dysfunction	2 (4%)	0 (0%)	0.072

AEs = adverse events. TEAEs = treatment-emergent adverse events.

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
