# Peer review of "Tadalafil 5 mg Alone or in Combination with Tamsulosin 0.4 mg for the Management of Men with Lower Urinary Tract Symptoms and Erectile Dysfunction: Results of a Prospective Observational Trial"

_jcm, 2019, doi:10.3390/jcm8081126_

Reviewer 1 Report

This paper reported the results of compare study of tadalafil 5 mg alone and combined with tamsulosin 0.4 mg for LUTS and ED patients. I cannot find any big problems in methods and results for this study. However, objects are not so remarkable and there are little new information obtained from this paper. I recommend making additional study plans from other new points of view, such as endothelial functions, metabolic syndromes.  

  1. I cannot understand why the authors set up “Run in 2 weeks tamsulosin 0.4 mg” before main comparison study? Of course, sequence of medications is important in this field. However, what is the meaning of this run in of tamsulosin? Do the authors plan any other sub analysis study based on these results?

 2. The authors described multicenter trial. But the data were obtained only from 2 urology department. In addition, the entry objects numbers are rather small in both groups for this kind of LUTS study.

 3. In title, Tamsulsin~ this is a huge misspelling, it is a title!

Author Response

This paper reported the results of compare study of tadalafil 5 mg alone and combined with tamsulosin 0.4 mg for LUTS and ED patients. I cannot find any big problems in methods and results for this study. However, objects are not so remarkable and there are little new information obtained from this paper. I recommend making additional study plans from other new points of view, such as endothelial functions, metabolic syndromes.  

 Dear Reviewer, Thank you for your comments. I agree with you that there are little new information; however, this is one the largest population of combination therapy (AB+PDE5-Is) published in current literature. I hope that, as suggested by you, this publication could encourage additional study on this topic.

  1. I cannot understand why the authors set up “Run in 2 weeks tamsulosin 0.4 mg” before main comparison study? Of course, sequence of medications is important in this field. However, what is the meaning of this run in of tamsulosin? Do the authors plan any other sub analysis study based on these results?

Thanks for your observation. Since alpha-blockers are considered both from GP and urologist one of the first line medication for men with LUTS and BPH, our aim was to simulate as much as possible the most commonly prescribed sequence of medications in daily clinical practice.

Therefore, after failure of first line monotherapy with tamsulosin, patients were switched to a pde5-Is as monotherapy, or a combination therapy of both medications.

2. The authors described multicenter trial. But the data were obtained only from 2 urology department. In addition, the entry objects numbers are rather small in both groups for this kind of LUTS study.

 I agree with you that the term  “multicenter” is misleading. We have removed this word from the text

3. In title, Tamsulsin~ this is a huge misspelling, it is a title!

I apologize for this misspelling. Now it’s correct

Reviewer 2 Report

This prospective placebo-controlled trial aimed to compare the efficacy and safety of 12-week daily administration of tadalafil 5 mg plus placebo with the combination of tadalafil 5 mg plus tamsulosin 0.4 mg in men with LUTS/BPH and ED. The results showed that Tadalafil monotherapy was able to improve ED, overall IPSS and Qmax at the end of the study. The combination of tamsulosin can further enhance the voiding symptoms and Qmax.

On the whole this is a well organized study and prepared manuscript. However there are some points that need to be addressed.

1. One of the main drawbacks of the study was not using randomized double-blind design. Moreover, the two groups of patients were recruited separately in two different hospitals. The lack of randomization could lead to bias in the study results.

2. The sample sizes in the two groups are different. The authors should explain the reason why and how they have remedied this discrepancy in the statistical analysis.

3.The case numbers in the two arms are relatively small. The calculation of study sample size and power should be provided.

4. The purpose of the 2-week tamsulosin run-in period should be explained.

Author Response

This prospective placebo-controlled trial aimed to compare the efficacy and safety of 12-week daily administration of tadalafil 5 mg plus placebo with the combination of tadalafil 5 mg plus tamsulosin 0.4 mg in men with LUTS/BPH and ED. The results showed that Tadalafil monotherapy was able to improve ED, overall IPSS and Qmax at the end of the study. The combination of tamsulosin can further enhance the voiding symptoms and Qmax.

On the whole this is a well organized study and prepared manuscript. However there are some points that need to be addressed.

1. One of the main drawbacks of the study was not using randomized double-blind design. Moreover, the two groups of patients were recruited separately in two different hospitals. The lack of randomization could lead to bias in the study results.

Thank you for your review. We agree with you, the design of the study is one of the main limitation of our study, as underlined in the discussion

2. The sample sizes in the two groups are different. The authors should explain the reason why and how they have remedied this discrepancy in the statistical analysis.

We focused on combination therapy: so we analyzed data with a 2:1 proportion.

3.The case numbers in the two arms are relatively small. The calculation of study sample size and power should be provided.

I agree with you that the two arms are small. However, in the present study there is one of the largest population available in current literature, regarding combination therapy with Abs + PDE5-Is.

4. The purpose of the 2-week tamsulosin run-in period should be explained.

Thanks for your observation. Since alpha-blockers are considered both from GP and urologist one of the first line medication for men with LUTS and BPH, our aim was to simulate as much as possible the most commonly prescribed sequence of medications in daily clinical practice.

Therefore, after failure of first line monotherapy with tamsulosin, patients were switched to a pde5-Is as monotherapy, or a combination therapy of both medications.

Reviewer 3 Report

It was not clear why the author gave run in 2 weeks of Tamsulosin and would not that affect the result?

It would be meaningful to add table that shows the improvement in sexual function from the base line with statistical analysis.

Fig 2 is not clear, it would be better if it get replaced with a table 

Table 3. results need statistical analysis.

In the conclusion the authors concluded that Tadalafil 5 mg daily monotherapy is able to improve ED and overall LUTS (IPSS and Qmax). although there has not been statistical analysis to compare it with the base line. 

Author Response

It was not clear why the author gave run in 2 weeks of Tamsulosin and would not that affect the result?

Thanks for your observation. Since alpha-blockers are considered both from GP and urologist one of the first line medication for men with LUTS and BPH, our aim was to simulate as much as possible the most commonly prescribed sequence of medications in daily clinical practice.

Therefore, after failure of first line monotherapy with tamsulosin, patients were switched to a pde5-Is as monotherapy, or a combination therapy of both medications.

It would be meaningful to add table that shows the improvement in sexual function from the base line with statistical analysis.

In table 2 are reported the differences from baseline and between the 2 groups

Fig 2 is not clear, it would be better if it get replaced with a table 

The results graphically highlighted in figure 2 are exposed in table 2. Based on your suggestion, now the figure is more impressive. We also changed the figure legend.

Table 3. results need statistical analysis.

According to your proposal, we add the p value in table 3

In the conclusion the authors concluded that Tadalafil 5 mg daily monotherapy is able to improve ED and overall LUTS (IPSS and Qmax). although there has not been statistical analysis to compare it with the base line. 

In table 2 the differences from baseline and between the 2 groups are reported.

Round  2

Reviewer 1 Report

This paper reported the results of compare study of tadalafil 5 mg alone and combined with tamsulosin 0.4 mg for LUTS and ED patients. As I mentioned in the first review, though I cannot find any big problems in methods and results for this study, objects are not so remarkable and there are little new information obtained from this paper.

However, the authors replied all the questions and amended almost adequately.